# Neutralizing monoclonal antibodies against the Gc fusion loop region of Crimean–Congo hemorrhagic fever virus

Liushuai Li[1,2◉], Tingting Chong[1,2◉], Lu Peng[1◉], Yajie Liu[1], Guibo Rao[1], Yan Fu[1], Yanni Shu[1], Jiamei Shen[1], Qinghong Xiao[1], Jia Liu[1], Jiang Li[1], Fei Deng[1], Bing Yan[1], Zhihong Hu[1], Sheng Cao[1]*, Manli Wang[1,3]*

1 State Key Laboratory of Virology, Wuhan Institute of Virology, Center for Biosafety Mega-Science, Chinese Academy of Sciences, Wuhan, China, 2 University of the Chinese Academy of Sciences, Beijing, China, 3 Hubei Jiangxia Laboratory, Wuhan, China

◉ These authors contributed equally to this work.
* caosheng@wh.iov.cn (SC); wangml@wh.iov.cn (MLW)

**Data Availability Statement:** The cryo-EM data is available at the following links: PDB DOI: https://doi.org/10.2210/pdb8JLX/pdb PDB DOI: https://

## Abstract

Crimean-Congo hemorrhagic fever virus (CCHFV) is a highly pathogenic tick-borne virus, prevalent in more than 30 countries worldwide. Human infection by this virus leads to severe illness, with an average case fatality of 40%. There is currently no approved vaccine or drug to treat the disease. Neutralizing antibodies are a promising approach to treat virus infectious diseases. This study generated 37 mouse-derived specific monoclonal antibodies against CCHFV Gc subunit. Neutralization assays using pseudotyped virus and authentic CCHFV identified Gc8, Gc13, and Gc35 as neutralizing antibodies. Among them, Gc13 had the highest neutralizing activity and binding affinity with CCHFV Gc. Consistently, Gc13, but not Gc8 or Gc35, showed *in vivo* protective efficacy (62.5% survival rate) against CCHFV infection in a lethal mouse infection model. Further characterization studies suggested that Gc8 and Gc13 may recognize a similar, linear epitope in domain II of CCHFV Gc, while Gc35 may recognize a different epitope in Gc. Cryo-electron microscopy of Gc-Fab complexes indicated that both Gc8 and Gc13 bind to the conserved fusion loop region and Gc13 had stronger interactions with sGc-trimers. This was supported by the ability of Gc13 to block CCHFV GP-mediated membrane fusion. Overall, this study provides new therapeutic strategies to treat CCHF and new insights into the interaction between antibodies with CCHFV Gc proteins.

## Author summary

Crimean-Congo hemorrhagic fever (CCHF) is a priority disease by the World Health Organization (WHO). It is a deadly viral infectious disease with case fatalities up to 40%, and no approved vaccine or treatment. Neutralizing antibodies are a promising approach toward treating viral infections, as exemplified in other hemorrhagic fevers such as Ebola. Currently, there are few reports of efficient neutralizing antibodies against CCHF virus

doi.org/10.2210/pdb8JKD/pdb PDB DOI: https://doi.org/10.2210/pdb8JLW/pdb.

**Funding:** This work was supported by grants from the National Key R&D Program of China (2022YFC2303300 to Z.H.; 2023YFC2305900 to M. W.; 2021YFF0702002 to J.L.), the National Natural Science Foundation of China (U22A20336 to Z.H.), Hubei Natural Science Foundation for Distinguished Young Scholars (2021CFA050 to M. W.), the Strategic Priority Research Program of the Chinese Academy of Sciences (XDB0490000 to Z. H.). The funders had no role in study design, data collection and analysis, decision to publish, or preparation of the manuscript.

**Competing interests:** M.W., L.L., Z.H., S.C., L.P., and Y.S. filed a patent for Gc13 in treatment of CCHF to China Intellectual Property Office. Remaining authors declare no conflicts of interests.

(CCHFV). This study successfully screened mouse-derived monoclonal antibodies with high *in vitro* neutralizing activity and *in vivo* protective efficacy against CCHFV infection. Among those, mAb Gc13 was the most efficient because it strongly bound to the highly conserved fusion loop regions of the CCHFV Gc subunit, thus blocking the crucial virus membrane fusion process. This study highlights the potential of neutralizing antibody-based strategies for CCHF treatment.

## Introduction

Crimean-Congo hemorrhagic fever (CCHF) is a globally widespread disease, particularly in Africa, Asia, southeastern Europe, and the Middle East. This disease may cause severe fever, leukopenia, and thrombocytopenia, a case fatality rate as high as 40%, and has been identified as a priority disease by the World Health Organization (WHO) [1,2]. The causative agent, Crimean-Congo hemorrhagic fever virus (CCHFV), is a highly pathogenic tick-borne virus belonging to the *Nairoviridae* family, *Bunyavirales* order, and must be handled in laboratories with high biosafety levels. The CCHFV genome contains three segments, designated small (S), medium (M), and large (L). The S segment encodes the nucleoprotein (NP) and nonstructural protein NS-**S** (NSs), and the M segment the glycoprotein precursor, which is further processed into the mucin-like domain (MLD), GP38, Gn, nonstructural protein M (NSm), and Gc protein. Structural proteins Gn and Gc are involved in viral attachment and entry [3]. Additionally, Gc is commonly used to generate neutralizing antibodies (nAbs). The L protein contains several motifs, including an ovarian tumor (OTU) cysteine protease [4], zinc finger, and RNA-dependent RNA polymerase (RdRp).

Supportive therapy is primary clinical therapeutic solution for CCHF treatment as there are no validated vaccines or treatment drugs for this infection. Certain compounds, such as favipiravir (T-705) [5,6], ribavirin [7] and the recently reported T-705-derived compound, H44 [8] can inhibit CCHFV infection *in vivo* and *in vitro*; however, the effect of ribavirin is controversial [9]. In addition to small-molecule drugs, neutralizing antibodies are a promising approach for the treatment of viral infections. The earliest neutralizing monoclonal antibodies (mAbs) targeting the CCHFV Gc and exhibiting protective effects in a mouse model were reported in 2005 [10]. In 2021, a panel of neutralizing mAbs targeting the CCHFV Gc was isolated from human survivors [11]. To enhance the antiviral efficacy of these mAbs, bispecific antibodies were generated by linking the variable domains of the two synergistic mAbs. Among these bispecific antibodies, DVD-121-801 (a combination of two neutralizing antibodies, ADI-36121 and ADI-37801) provided complete protection against CCHFV challenge in a lethal mouse infection model. The binding of ADI-36121 to domain II enables the fusion loop to be more accessible to ADI-37801, which provides structural insights into the mechanism of bispecific antibodies and elucidates further engineering and improvement of nAbs [12].

In this study, we generated 37 mouse-derived mAbs against the CCHFV glycoprotein Gc subunit, and three nAbs: Gc8, Gc13, and Gc35, were selected for further studies. Gc13 showed the highest neutralizing activity, with a median inhibitory concentration ($IC_{50}$) of ~40 ng/mL *in vitro*, and exhibited a protective effect in CCHFV-challenged mice. The cyro-EM structures of Gc-Fab complexes and fusion inhibition assays showed that Gc8 and Gc13 could bind to the fusion loops of Gc and inhibit membrane fusion.

## Results

### Neutralizing antibody screening by pseudotyped CCHFV

Mice were immunized with sGc-trimer proteins of CCHFV IbAr10200 strain via subcutaneous injection, and further boosted twice at two-week intervals. Then, the splenocytes were fused with SP/20 myeloma cells. Enzyme linked immunoassay (ELISA) was used for monoclonal screening, and 37 hybridomas positive for CCHFV Gc proteins were selected. To preliminarily evaluate the neutralizing activities of mAbs generated by these hybridomas, we established a VSV-based pseudotyped virus with CCHFV GP (VSVΔG/GFP-CCHFV GP) and conducted a pseudotyped virus neutralizing assay (PNA) following the screening procedure illustrated in **Fig 1A**. Three antibody dilutions (1:10, 1:40, and 1:160) were tested and showed dose-dependent neutralizing effects for most antibodies. Among these, 25 mAbs with inhibitory ratios exceeding 50% (**Fig 1B**) at the dilution of 1:160 were selected and the corresponding hybridomas were used to immunize mice, and mAbs were purified from mouse ascites for further

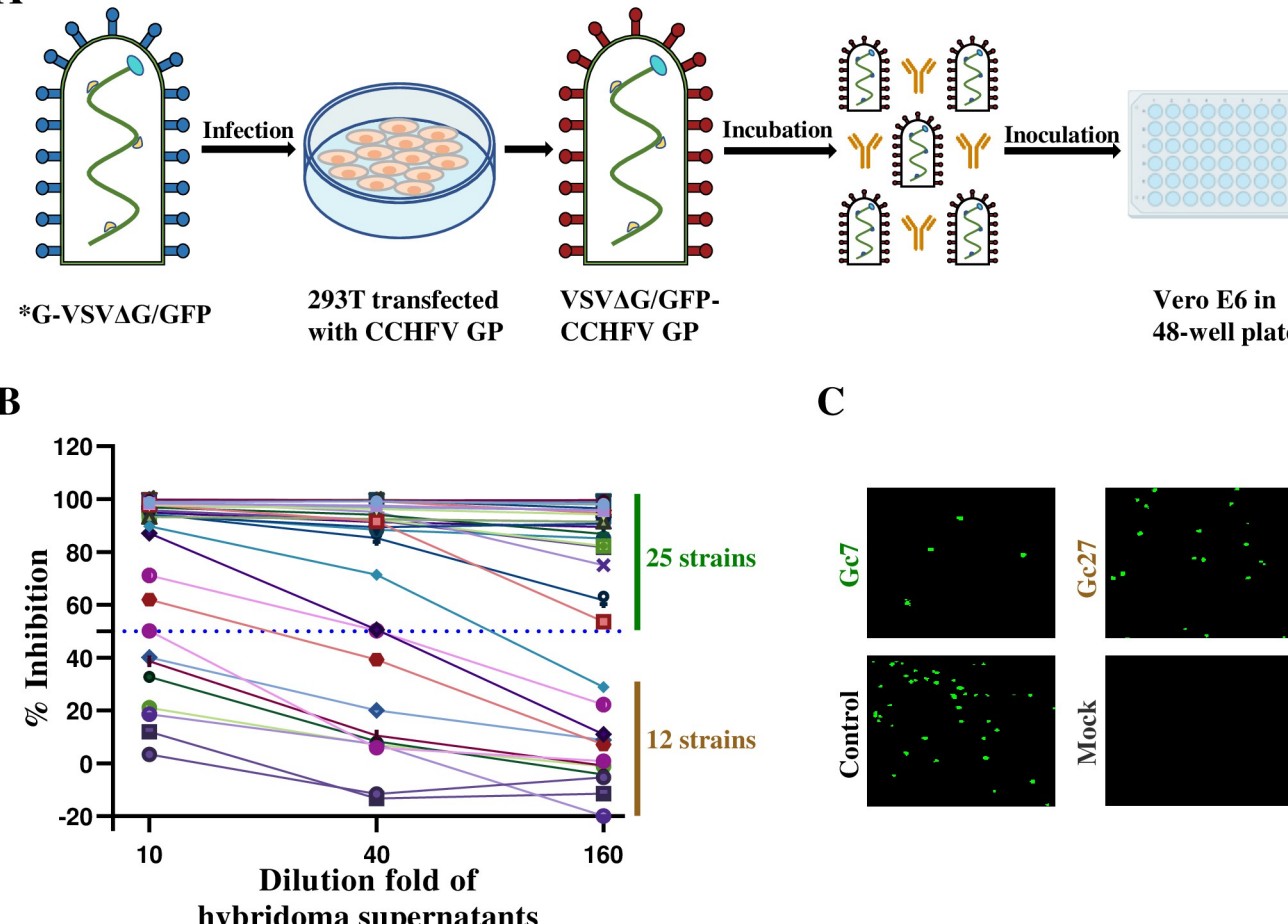

**Fig 1. Preliminary screening of neutralizing antibodies.** (A). Flow chart of nAb screening by CCHFV pseudotyped virus. *G-VSVΔG/GFP was used to infect HEK293T cells that were pre-transfected with CCHFV GP 24 h earlier. After incubation at 37°C for 24 h, the supernatant containing pseudotyped CCHFV was collected. For neutralization assay, 3000 TCID$_{50}$ pseudotyped CCHFV was mixed with mAbs at 37°C for 1 h, and the mixture was added to Vero E6 cells for 1 h. After an additional 24 h incubation at 37°C, GFP expressing cells were counted under a fluorescence microscope. (B). Thirty-seven mAbs with 10, 40, and 160 dilution fold were tested for neutralization effects against pseudotyped CCHFV as mentioned above. Twenty-five mAbs could inhibit > 50% pseudotyped CCHFV infection at the dilution of 1:160. (C) Representative images at the antibody dilution of 1:160.

experiments (**Fig 1B**). Representative fluorescence images from each group are shown in **Fig 1C**.

## Neutralizing activity of Gc mAbs against live CCHFV *in vitro*

Subsequently, we evaluated the neutralizing activity of the 25 selected mAbs against live CCHFV-YL16070 using a microneutralization assay in Vero E6 cells. After the initial neutralizing assay using an antibody concentration of 100ng/mL, Gc13, Gc35, and Gc8 exhibited higher neutralizing activities than the other mAbs. Therefore, we selected the three antibodies above to determine their $IC_{50}$. Summarily, three-fold serially diluted antibodies were used for the microneutralization assay against the CCHFV-YL16070 live virus. At 48 h p.i., the cell supernatants were used to quantify viral genomic copies by qRT-PCR to calculate the $IC_{50}$. The calculated $IC_{50}$ values of Gc8 and Gc35 against live CCHFV were 484.9 ng/mL and 353.9 ng/mL, respectively. While for Gc13, the $IC_{50}$ value was 40.2 ng/mL, which was almost 10-fold lower than that of the other two mAbs (**Fig 2A**). Quantification of the released infectious progenies in infected supernatants determined the $IC_{50}$ values of Gc8, Gc13, and Gc35 against CCHFV to be 591.2, 54.5, and 357.8 ng/mL, respectively (**Fig 2B**). Therefore, of the three mAbs tested, Gc13 exhibited the highest neutralizing activity.

## Gc8 and Gc13 have a protective effect against CCHFV challenge in mice

Animal experiments were conducted to test the *in vivo* efficacy of the three mAbs against CCHFV-YL16070 using a previously reported lethal mouse model [8]. Notably, none of the

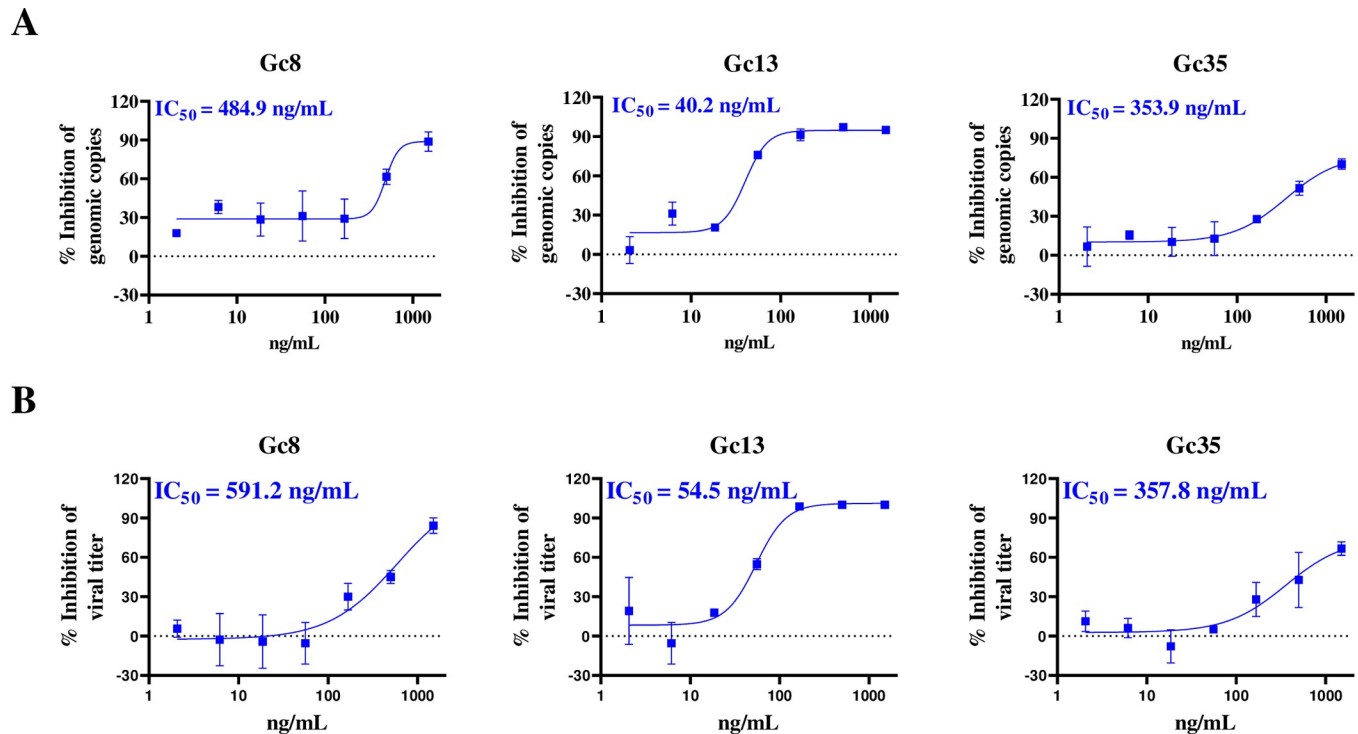

**Fig 2. Neutralizing activities of Gc mAbs against live CCHFV-YL16070 *in vitro*.** $IC_{50}$ values of Gc8, Gc13, and Gc35 against live CCHFV-YL16070 at an MOI of 0.01. After incubation with serially diluted mAbs, the virus-antibody mixture was used to infect Vero E6 cells. Three days later, the viral copies in supernatant were detected by qRT-PCR (A) and the infectious progeny in the supernatant was determined using end-point dilution assay (B). The inhibition rates to DMSO control were calculated and graphed using GraphPad Prism. The experiments were performed in triplicates.

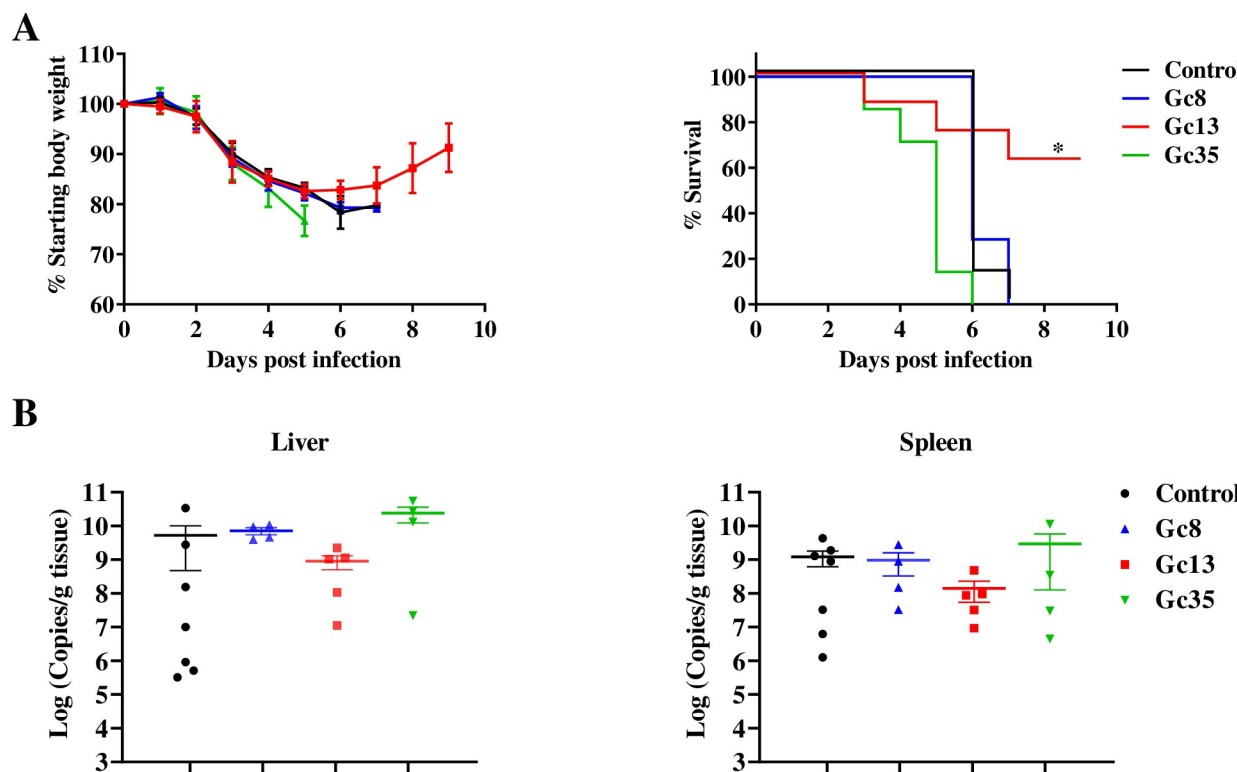

**Fig 3. *In vivo* protective efficacy of Gc mAbs against CCHFV challenge.** (A). Weight loss and survival curves of infected mice treated with Gc8, Gc13 and Gc35. The C57BL/6J-IFNAR$^{-/-}$ mice (7–8 mice per group) were pre-treated with 50 mg/kg mAbs or equivalent volume of PBS control via intraperitoneal route. After 24 h, mice were infected with 5,000 TCID$_{50}$ of CCHFV-YL16070 via the intraperitoneal route. Weight loss and clinical symptoms were monitored daily. Statistical analysis was performed using Simple survival analysis (Kaplan-Meier). *$P < 0.05$. (B). Viral copies in the liver and spleen of infected mice. Total RNA in the infected tissues were extracted and transcribed into cDNA. Viral RNA was quantified by qRT-PCR and transformed into RNA copies per gram of tissues.

three mAb (50 mg/kg) showed any impact on the body weight change and the survival of animals in the absence of CCHFV infection (**S1 Fig**). Upon CCHFV infection, as shown in **Fig 3A**, body weight loss in the control group began as early as 2 days post infection (d p.i.). Between 6–7 d p.i., all the mice in the control group either died of infection or reached the ethical endpoint (8/8; 100% mortality rate). Following the administration of a single 50 mg/kg dose of each mAb 24 h before viral infection, Gc13 improved the survival rate of CCHFV infected mice to 62.5% (5/8) ($P < 0.05$), with 2 (2/8) mice dying at 3 and 5 d p.i. and 1 (1/8) mouse euthanized at 7 d p.i. Mice treated with Gc8 and Gc35 showed no difference in survival rates compared with the control group.

For Gc13, although there was no statistically significant difference, viral loads in the spleen and liver were reduced by approximately one log, compared to those in the control group, whereas treatment with Gc8 or Gc35 had no inhibitory effect on viral replication in the spleen and liver of infected mice (**Fig 3B**). Therefore, Gc13 showed protective efficacy against lethal CCHFV infection *in vivo*.

## Gc8, Gc13 and Gc35 can recognize native Gc proteins in virus-infected cells

To investigate whether these three nAbs could recognize Gc proteins in CCHFV-infected cells, western blot, and immunofluorescence staining were performed. As shown in **Fig 4A**, all three mAbs recognized Gc proteins according to western blot analysis. A specific protein band with

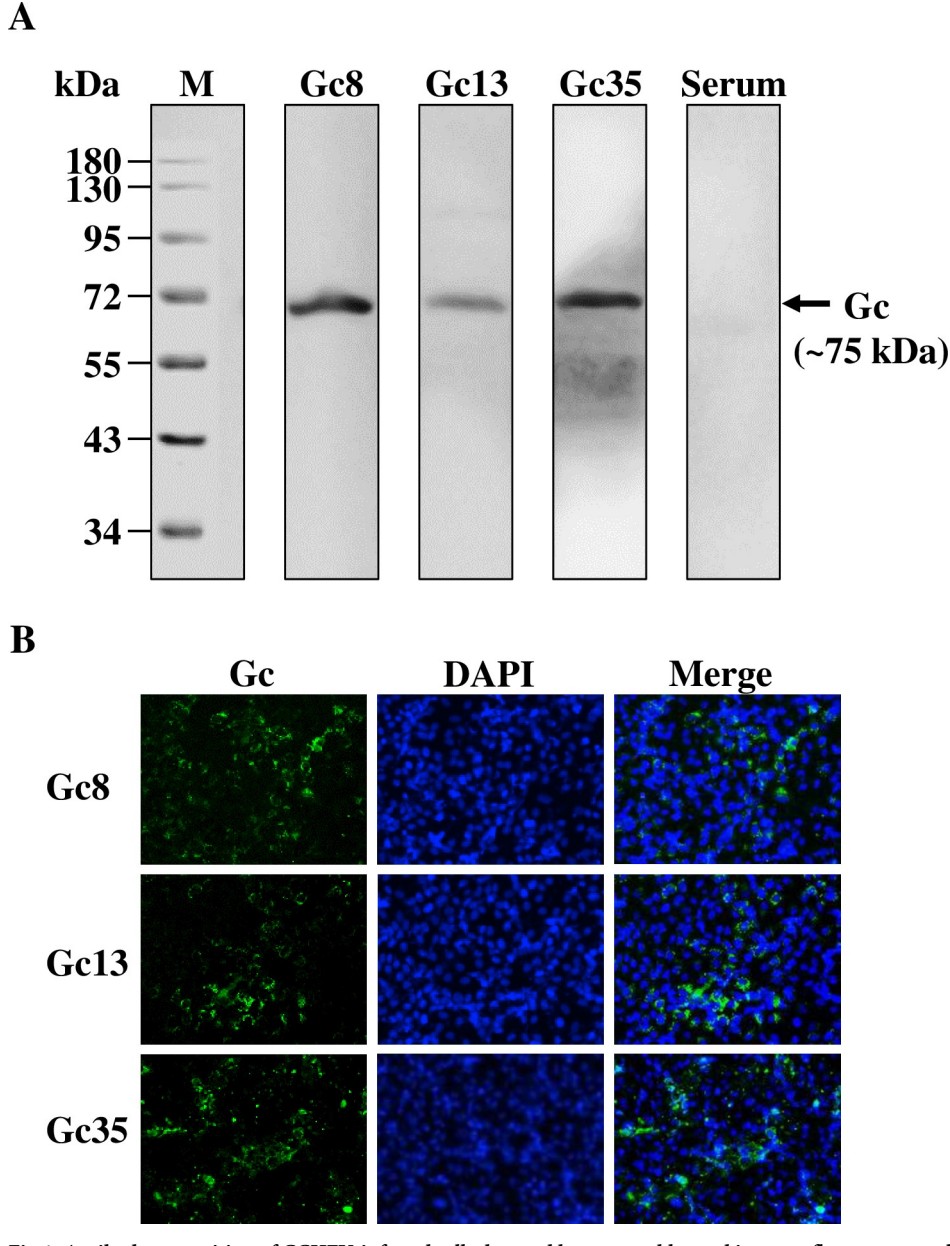

**Fig 4. Antibody recognition of CCHFV-infected cells detected by western blot and immunofluorescent analysis.** (A). Protein samples of CCHFV-YL16070 infected Vero E6 cells were subjected to western blot analysis. Gc8, Gc13, Gc35, and negative mouse serum were used as primary antibodies at a dilution of 1:5,000. The bands, approximately 75 kDa, indicated by the black arrow suggested that all 3 mAbs could recognize linear epitopes of Gc. (B). Vero E6 cells were infected with CCHFV-YL16070 at an MOI of 1. At 72 h p.i., samples were collected and subjected to indirect immunofluorescence assay using Gc8, Gc13, and Gc35 as the primary antibodies at a dilution of 1:5,000. Fluorescent signals represented Gc proteins detected by the three mAbs.

a molecular weight of approximately 75 kDa was detected, which corresponded to the size of mature Gc proteins [13]. These results suggest that Gc8, Gc13, and Gc35 recognize the linear epitopes in Gc.

To determine whether Gc8, Gc13, and Gc35 can recognize native Gc proteins, immunofluorescence staining was performed using CCHFV-infected Vero E6 cells in 96-well plates. As shown in **Fig 4B**, a fluorescent signal was detected in cells infected with all three mAbs. The

Gc proteins detected by Gc8, Gc13, and Gc35 were mainly located in the cytoplasm of infected cells. These data suggest that Gc8, Gc13, and Gc35 recognize native Gc proteins during viral infection.

## Binding characteristics of Gc8, Gc13, Gc35 mAbs

To determine the binding characteristics of Gc8, Gc13, and Gc35, competitive ELISA was performed as described in a previous study [14]. As shown in **Fig 5A**, Gc8 and Gc13 competed with each other, as evidenced by a dose-dependent decrease in OD values. In contrast, Gc35 did not compete with Gc8 or Gc35. Subsequently, to determine the region in the Gc protein recognized by Gc8, Gc13, and Gc35, 4 truncated Gc protein fragments (aa 1–142, 132–262, 252–375, and 365–521), each containing 10 overlapping residues, were expressed in *E.coli*. The four truncated Gc proteins were probed using primary antibodies against Gc8, Gc13, and Gc35. Western blot analysis showed that Gc8 and Gc13 recognized the same truncated region (aa 132–262, mainly domain II) in Gc, whereas Gc35 recognized aa 365–521 (mainly domain III) at the C-terminus of the Gc ectodomain (**Fig 5B**). Therefore, the western blot and ELISA results were consistent.

As Gc8 and Gc13 recognized a similar region in Gc but their neutralizing abilities (IC$_{50}$ values) differed (**Fig 2A**), we performed a BLI experiment to compare the binding affinities of the three mAbs to Gc proteins [14]. As shown in **Fig 5C**, Gc13 showed the lowest KD value ($3.24 \times 10^{-11}$ M), representing the highest binding affinity to the Gc protein. Gc8 and Gc35 showed similar KD values ($8.34 \times 10^{-11}$ M and $8.79 \times 10^{-11}$ M, respectively), which validated the results of the neutralization assay (**Fig 2A**).

## Gc8 and Gc13 block CCHFV GP-mediated membrane fusion

Because fusion loops, the key regions for mediating the membrane fusion process, are localized in the segment targeted by Gc8 and Gc13, we tested the ability of the three mAbs to inhibit membrane fusion. As illustrated in **Fig 6A**, a plasmid encoding eGFP under the control of the T7 promoter was transfected into Huh-7 cells, and another plasmid encoding CCHFV GP was transfected into BSR-T7/5 cells stably expressing T7 RNA polymerase. After being triggered with low-pH medium, these two cells would amalgamate, mediated by the Gc protein expressed on the plasma membrane of BSR-T7/5 cells, and then the T7-eGFP plasmid would transcribe the eGFP by T7 RNA polymerase expressed in BSR-T7/5 cells. If the nAbs could inhibit CCHFV-GP-mediated membrane fusion, these two cell types would not amalgamate; therefore, no syncytium formation with eGFP signals would be observed. As shown in **Fig 6B**, Gc13 could inhibit CCHFV-GP-mediated membrane fusion by > 70% at a concentration of 100 ng/mL ($P < 0.001$), whereas Gc8 significantly inhibited syncytium formation at 100 ng/mL (~ 35%) significantly ($P < 0.001$). In contrast, Gc35 did not inhibit CCHFV GP-mediated synthesis. The representative images at 100 ng/mL are shown in **Fig 6C** and the edges of syncytia are outlined using a white line. These results suggest that Gc8/13 inhibits CCHFV infection by blocking GP-mediated membrane fusion.

## Structural characterization of the sGc-trimer in complex with multiple Fab fragments

To study the molecular mechanisms underlying Gc8 or Gc13 mediated neutralization, we characterized the complex structures of CCHFV sGc-trimers and Gc8 or Gc13 Fabs using transmission electron microscopy (TEM). Negative staining TEM of Fab–sGc-trimer complexes yielded extra density compared to sGc-trimers (**S2 Fig**), suggesting that Gc8 or Gc13 Fab interacted with sGc-trimers to form larger complexes. The 2D class averages of Fab–sGc-

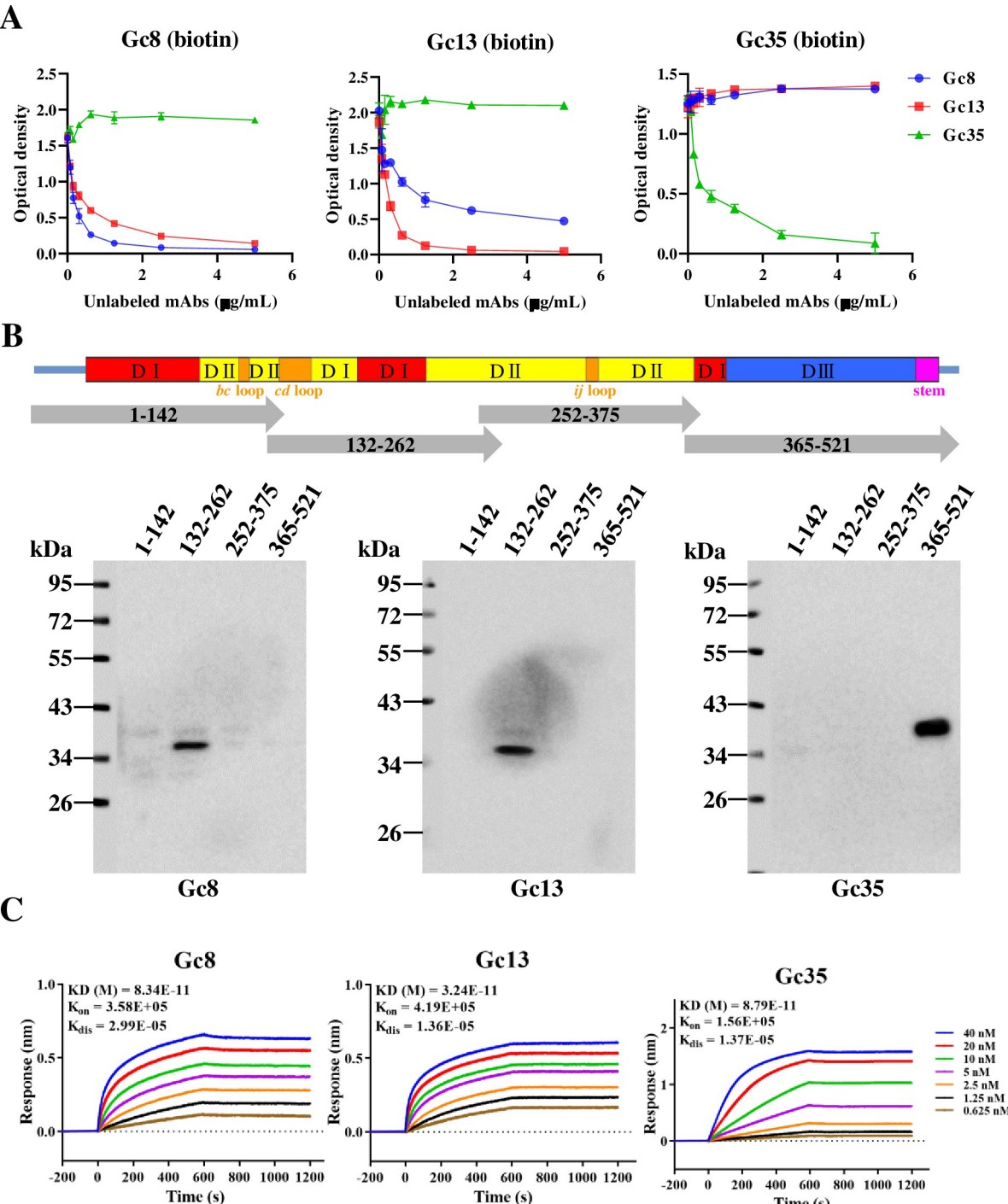

**Fig 5. Binding characteristics of Gc8, Gc13, and Gc35.** (A). Competitive ELISA of Gc mAbs for epitope determination. Biotin-labeled antibodies and serially-diluted unlabeled antibodies were mixed and added to sGc-S-tag protein-coated plates for ELISA and the OD values were determined using a microplate reader. The results showed that Gc8 and Gc13 could compete with each other as the OD values decreased by increasing another antibody concentration. (B). The recognizing regions in Gc proteins by mAb were determined using western blot analysis. Different domains (DI, DII, DIII, Stem region, and fusion loops) of Gc were classified according to the Gc structure information (PDB: 7FGF) and shown in different colors. Four truncated Gc fragments (gray bars) were cloned into pET32a vector and protein expression was induced by IPTG. Protein samples were collected and subjected to western blot using the mAbs as primary antibodies at a dilution of 1:5,000. (C). Binding affinity assay of Gc8, Gc13, and Gc35 to sGc-S-tag by BLI. The binding curves and kinetics of association and dissociation were analyzed using ForteBio Data Analysis Software.

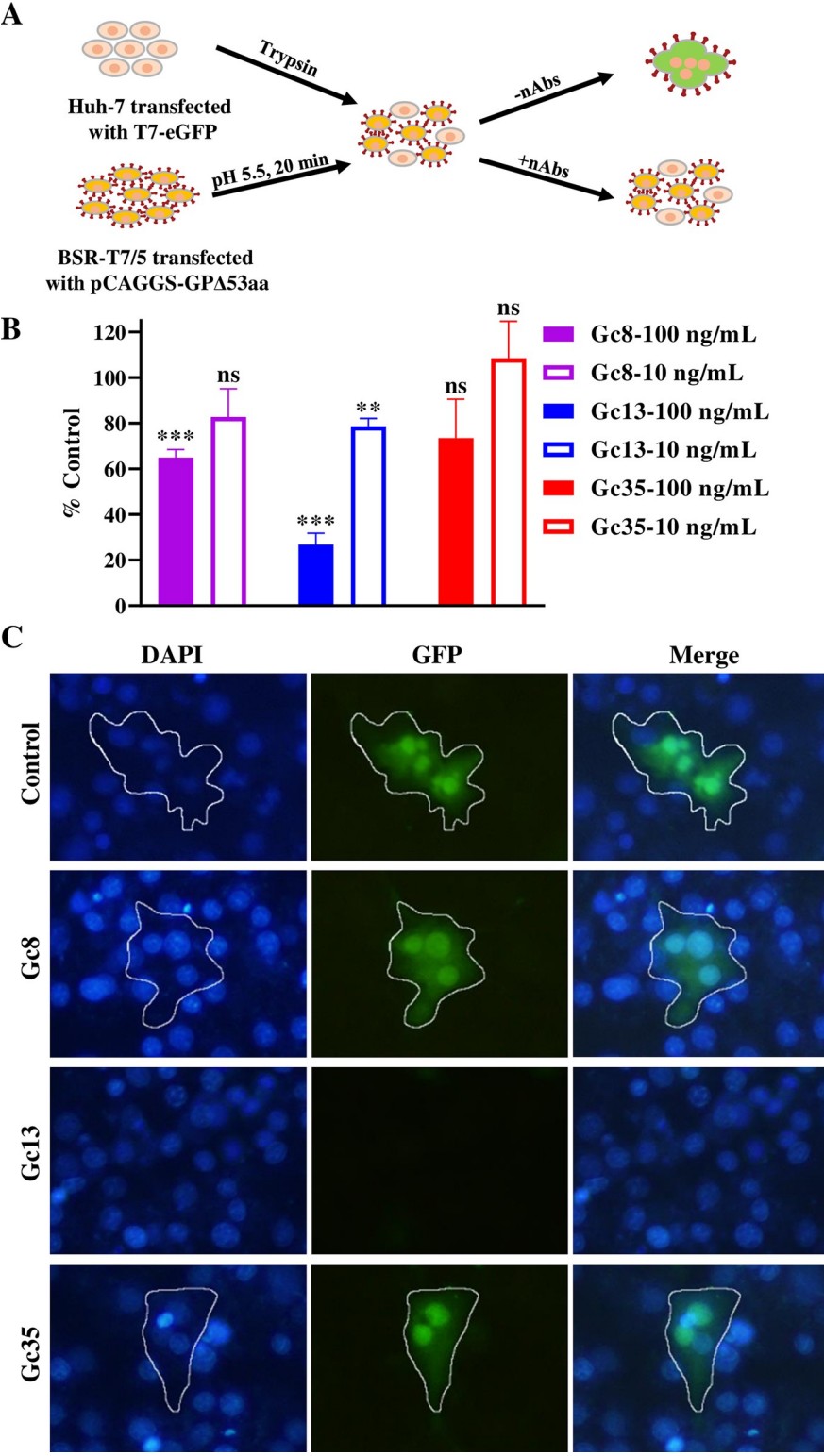

**Fig 6. Gc8 and Gc13 block CCHFV GP-mediated membrane fusion.** (A). Flow chart for fusion inhibition assay. Huh-7 cells were transfected with a pT7-eGFP plasmids and mixed with BSR-T7/5 cells transfected with pCAGGS-CCHFV-GPdel53aa. After induction by low-pH medium, these cells would fuse with each other to form syncytia with GFP expression, and the addition of nAbs, but not medium control, would inhibit this process. (B). Quantification of syncytium inhibition rate of Gc8, Gc13, and Gc35 at 10 and 100 ng/mL. Statistical analysis was

performed using unpaired T-test. **$P < 0.01$, ***$P < 0.001$. (C). Representative fluorescent images of each nAb at 100 ng/mL were shown and the edges of syncytia are outlined using a white line.

trimer complexes in cryo-EM micrographs showed that one or two Fab molecules interacted with a CCHFV sGc-trimer at the distal end of the trimers (**Fig 7A and 7B, arrows**). Heterogeneous refinement enabled the identification of the complexes when one, two, or three Fabs

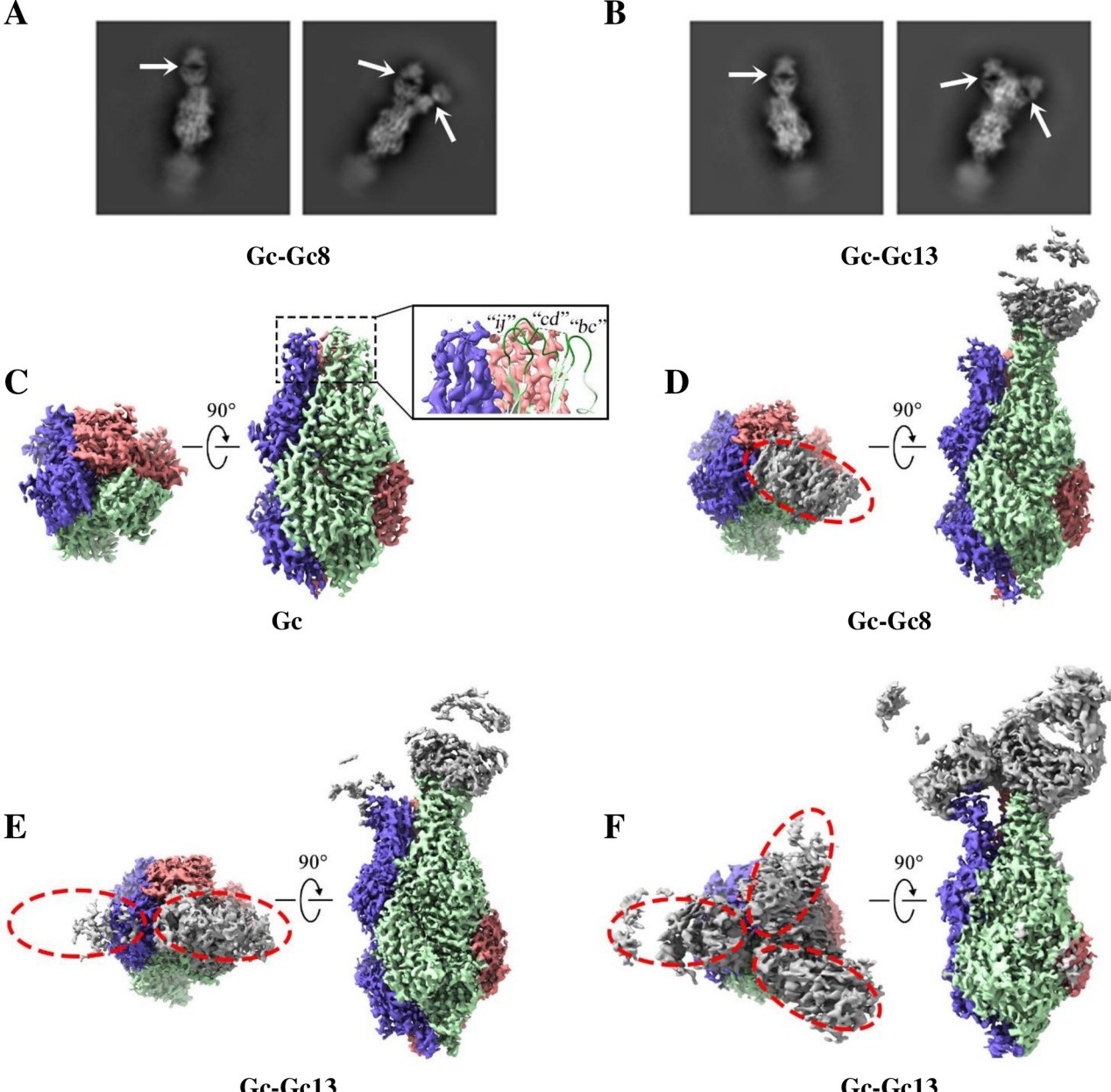

**Fig 7. Binding stoichiometry between sGc-trimers and Fab fragments.** (A). Representative 2D class averages showing that the sGc-trimer can bind to one Gc8 Fab (left) or simultaneously interact with two Gc8 Fabs (right). The arrows indicate the bound Fab molecules. (B) Representative 2D class averages of sGc-trimer in complex with one Gc13 (left) or two Gc13 Fab (right). (C) Orthogonal views of molecular surface representation for sGc-trimers without bound Fab (EMD-31579). The three sGc protomers are presented in red, purple and green. The three fusion loops from the green sGc molecule are presented in cartoon form in the inset. (D–F) Orthogonal views of the sGc-trimer in complex with a single Gc8 Fab (D), two Gc13 Fab, (E) and three Gc13 Fab (F). Density corresponding to Fabs was presented in gray and roughly outlined by elliptical dashed lines.

were bound to the Gc-trimer (**Fig 7D–7F, red circles**). Compared to Gc8, more Gc13 bound to sGc-trimers, with 59.5% complex (Class 3 in **S4 Fig**), showing the density for the third Gc13 Fab, whereas only two Gc8 Fabs could bind to the trimers (Class 3 in **S3 Fig**). However, when two or three Fabs bind to sGc trimers, the electron density of the second or third Fab is attenuated in the cryo-EM map, largely owing to the steric conflict between closely packed Fabs. To solve the Fab–Gc interface at a higher resolution, sGc trimers with relatively few bound Fabs were selected for further non-uniform (NU) refinement. The Fab–Gc complex structures generally identified at approximately 2.6-Å resolution reveal that the fusion loop regions are epitopes targeted by both Gc8 and Gc13.

## Antibody-antigen interactions

Localized reconstructions focused on the interface between Fab and Gc provided more optimized interpretable density (at approximately 3 Å resolution) for fusion loops of the Gc-trimer and also complementarity-determining regions (CDRs) of the variable domains of Fabs. The Fab–Gc interaction buries a surface area of 670 Å$^2$ or 778 Å$^2$ on Gc by 6 CDR loops of the Gc8 or Gc13 Fab, respectively, through hydrophobic contacts and hydrogen bonding. The intermolecular interactions (**S2**–**S6 Tables**) are mainly contributed by the CDRH loops of Gc8 or Gc13 and the *cd* loop of Gc, which are disordered in the structure of Gc-trimers without Fabs [15]. W1191 and W1199 from the Gc "*cd*" loop form hydrophobic core with hydrophobic residues from CDRHs of Gc8, while the hydrophobic core was formed by W1197 and W1199 from the "*cd*" loop and residues from CDRHs of Gc13 (**S2**–**S6 Tables**). In addition, based on online PISA analysis (https://www.ebi.ac.uk/pdbe/pisa/), two hydrogen bonds were identified at the Gc8–Gc interface, whereas nine were identified at the Gc13–Gc interface (**Fig 8A and 8B, and S6 Table**). In the Gc8-Gc complex, C1165 and T1196 on the fusion loops form hydrogen bonds with Gc8 heavy chain Y102 and light chain Y92, respectively (**Fig 8A and S6 Table**). Among the nine hydrogen bonds at the Gc13-Gc interface, seven are formed by the residues from the "*cd*" loop. W1197 and W1199 mentioned above, critical for forming the hydrophobic core, are also involved in forming two hydrogen bonds. In addition, the main-chain carboxyl oxygen atoms of A1163 near the "*bc*" loop and G1363 from the "*ij*" loop contribute to two more hydrogen bonds (**Fig 8B and S6 Table**). The larger interface between Gc13 and Gc and more intermolecular hydrogen bonds indicate a stronger binding affinity of Gc13 to the fusion loops of Gc, which is consistent with the observation that more Gc13 Fabs could bind to sGc-trimers, despite the structural collision among bound Fabs, compared to the Gc8 Fabs. Interestingly, the binding of Gc8 and Gc13 to Gc is essentially in the opposite orientation, resulting in the VH domain of Gc8 being well aligned with the VL domain of Gc13, and the VL of Gc8 being superposable to the VH of Gc13 (**Fig 8C**). Compared to the Gc13 Fab, another reported Fab (ADI-37801) formed 4 hydrogen bonds with the *cd* loop, but with different binding orientations (**Fig 8D**).

## Discussion

In previous research, a panel of mAbs produced from mice immunized with infected suckling mouse brain homogenates or affinity-purified glycoproteins using hybridoma technology exhibited broad neutralizing activity *in vitro* against different strains of CCHFV [10,16]. A recent study isolated a panel of mAbs from convalescent donors in Uganda using a recombinant protein bait (rGn/Gc) containing the ectodomains of Gn and Gc [11]. Interestingly, most of these mAbs (> 80%), such as ADI-37801 and ADI-42404, target CCHFV Gc. Based on above information, we selected the ectodomain region of Gc as the immunogen to immunize mice and generate mAbs. Through evaluation using the pseudotyped virus and live CCHFV,

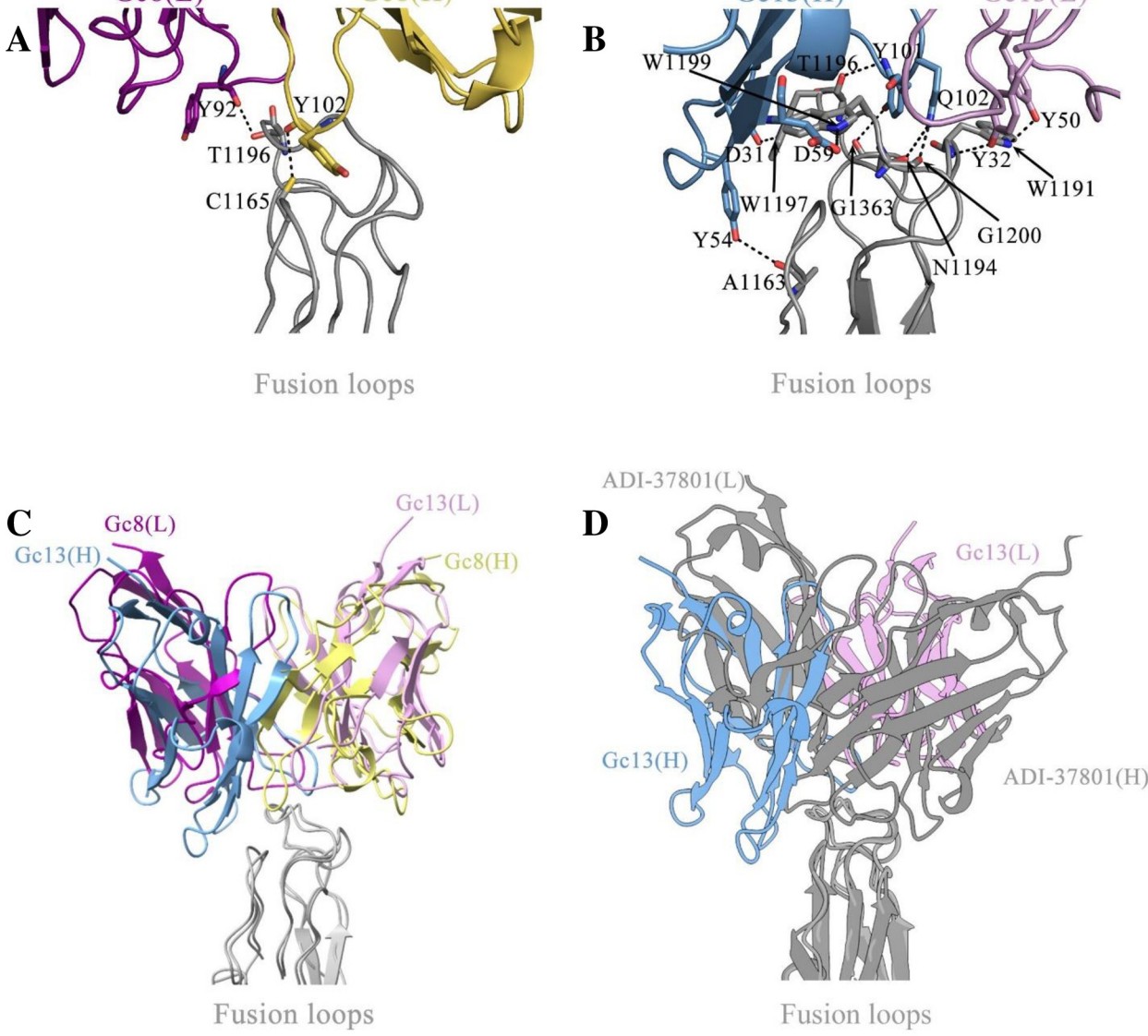

**Fig 8. Interactions of Gc8/13 Fabs with CCHFV Gc.** (A–B) Close-up view of the hydrogen bonding networks between fusion loops of CCHFV Gc and Gc8 (A) or Gc13 (B). Variable heavy and light (VH and VL) chains are presented in distinct colors for both Gc8 and Gc13. (C) Fab–Gc complexes, focusing on their interface, are superimposed on fusion loops of Gc. Color codes are similar to those in (A) and (B). (D) Structural comparison between Gc13–Gc and ADI-37801 complexes. Fusion loops are superimposed.

three nAbs, designated as Gc8/13/35, were sorted out and their *in vitro* neutralizing ability, *in vivo* efficacy, and mode of action were analyzed.

We have compared the neutralizing abilities of some previously reported nAbs with the ones generated in this study. CCHFV Gc-targeted mAbs obtained from mice could effectively neutralize CCHFV-IbAr10200 strain *in vitro* with the 50% plaque reducing neutralizing titers (PRNT$_{50}$) of < 1 µg/mL (0.18 µg/mL for 8A1, 0.31 µg/mL for 30F7, and 0.33 µg/mL for 11E7) [16,17]; nAb isolated from human plasma could effectively neutralize CCHFV-IbAr10200 strain *in vitro* with IC$_{50}$ ranging from less than 0.01 nM (1.5 ng/mL, ADI-36121) to approximately 1 nM (150 ng/mL, ADI-37801) [11]. Neutralizing mAbs in our study could significantly neutralize CCHFV-YL16070 strain *in vitro* with IC$_{50}$ values of 40.2 ng/mL (Gc13), 353.9 ng/

mL (Gc35), and 484.9 ng/mL (Gc8) (**Fig 2A**). *In vivo* study showed that mAbs derived from mice (8A1 and 11E7), although with low $IC_{50}$ values, failed to protect mice challenged with CCHFV [18]. In our study, Gc13 showed an *in vivo* protective effect (62.5% survival rate) when administered 1 day before viral infection (**Fig 3A**). Human plasma derived mAbs (ADI-36121, ADI-36145, and ADI-37801) showed basically complete protection with prevention of mouse body weight loss when administered 1 day prior to CCHFV challenge, and the bispecific antibody, DVD-121-801, could offer partial protective efficacy (80% survival rate) when administered 1 day post infection. The *in vivo* protective efficacy of nAbs may be related to their *in vitro* neutralizing activity (the lower the $IC_{50}$ value, the better *in vivo* protective efficacy) to a certain extent. However, this can also be co-determined by other factors such as different antibody generation strategies and antibody-binding epitopes, *etc*. For example, nAbs derived from CCHF-convalescent plasma may yield more optimized *in vivo* protective efficacy against CCHFV compared to mouse-originated nAbs as evidenced by less *in vitro* neutralizing potency ($IC_{50}$ = 150 ng/mL for ADI-37801 versus $IC_{50}$ = 40 ng/mL for Gc13) but higher *in vivo* protective efficacy of ADI-37801 compared to Gc13. The native viral glycoproteins in authentic virus would possibly elicit nAb with more optimized protective efficacy *in vivo*, compared to using recombinant Gc proteins as the immunogen. In addition, different animal infection models (animal types/viral strain/viral challenge dose, etc.) used for mAb evaluation may influence results, thus complicating direct comparison. For example, we used 5,000 $TCID_{50}$ of CCHFV YL16070 strain to challenge mice, while the parameters were applicable for 100 PFU/IbAr10200 for human nAbs evaluation [11].

Functional analyses, particularly our cryo-EM studies, confirmed the presence of Gc8 and 13 target fusion loops in domain II of Gc (**Fig 7**). The highly flexible *cd* loop (missing density in the antibody-free sGc-trimer) is stabilized upon the binding of Fabs. The *cd* loop at the tip of the post-fusion Gc-trimer may still be sufficiently flexible to allow for some local structural variation and thus accommodate more than one Fab molecule despite clashes between neighboring Fabs, while the binding orientation could be slightly rotated (**Fig 7E and 7F**). Gc13 showed stronger Fab-Gc interactions than Gc8 (**Fig 8**), which is consistent with the observation that more Gc13 Fabs bind to sGc-trimers (**Fig 7**). Therefore, the structural information provided a possible explanation for the higher binding affinity (**Fig 5C**) and neutralizing activity (**Fig 2A**) of Gc13 compared to those of Gc8. Interestingly, in the study of human derived mAbs, numerous mAbs (173/361) were identified against site 1, which encompasses the fusion loop region of Gc [11], suggesting that the site is a dominant nAb epitope.

CCHFV was highly variable in its genome sequences, with nucleotide acid divergence of 22, 31, and 20% and amino acid divergence of 10, 27, and 8% among the S, M, and L segments [19,20]. In contrast, the fusion loop region, which was targeted by Gc8 and Gc13, as well as the previously reported ADI-37801 [12], was highly conserved among different isolates from all 8 phylogenetic clades (**S5 Fig** and **S7 Table**). *Bc*, *cd*, and *ij* loops showed > 99% amino acid identity. Among the more than 100 analyzed isolates (only 33 selected sequences are presented), only 4 viral isolates showed a single amino acid variation in *bc* (one isolate), *cd* (two isolates), and *ij* (one isolate) loops (**S7 Table**). This suggests that Gc8 and Gc13 may possess broad neutralization activities across different CCHFV strains, similar to that observed in ADI-37801 [11]. In our study, the sGc-trimer proteins used to generate mAbs belonged to CCHFV-IbAr10200 (lineage Africa 2, **S5 Fig**), whereas the live virus-neutralizing assay and *in vivo* efficacy evaluation used CCHFV-YL16070 (lineage Asia 3; **S5 Fig**), thus highlighting their potential as broadly active nAbs.

However, we also realized that the *in vivo* protective efficacy of Gc13 was moderate. In fact, nAbs derived from human survivors also exhibited incomplete protection when nAbs were evaluated therapeutically, whereas DVD-121-801, a bsAb combining variable domains of

ADI-36121 and ADI-37801, offered partial (~80%) therapeutically protective efficacy. Therefore, further strategies to produce nAbs with higher neutralization activity and generate bispecific antibodies or nAb cocktails using Gc13 with other synergistic nAbs should be tested. In addition, certain non-neutralizing mAbs targeting the CCHFV GP38 domain could also protect mice from lethal infection [18,21,22], and are therefore worth considering for CCHF treatment, either used alone or combined with nAbs.

Notably, antibody-based therapy in other hemorrhagic fevers such as Ebola has yielded positive results in clinical trials. For example, triple monoclonal antibody REGN-EB3 containing a cocktail of three mouse derived mAbs (REGN 3470–REGN 3471–REGN 3479) and a single monoclonal antibody MAb114 isolated from human survivors showed remarkable efficacy in a randomized, controlled trial for which 681 patients in the Democratic Republic of the Congo (DRC) were enrolled. Both REGN-EB3 and MAb114 were formally approved by FDA for treating the Ebola virus in 2020. Treatment in the early phase of infection when the viral load is low (nucleoprotein Ct value > 22) would significantly reduce mortality, highlighting the advantage of early administration of mAbs in combating viral infectious disease [23–25]. Notably, both REGN-EB3 and MAb114 showed ideal therapeutic efficacy in treating the Ebola virus infection in animal models [23,24]. Therefore, the development of more efficient CCHFV nAbs is necessary. Although antibody-based therapy of CCHFV has yet to be tested in clinical trials, related research is rapidly growing. Given the successful application of mAbs in the treatment of the Ebola virus, we expect promising advancement in their use in treating CCHF in future.

## Methods

### Cells and viruses

Vero E6 cells (American Type Culture Collection [ATCC], no. 1586) were maintained in Eagle's minimum essential medium (EMEM, Gibco, Grand Island, NY, USA) supplemented with 10% fetal bovine serum (FBS; Gibco, Grand Island, NY, USA). HEK 293T (ATCC; no. CRL 3216), Huh-7 (National Collection of Authenticated Cell Cultures, Shanghai, China; TCHu182), and BSR-T7 cells (a kind gift from Dr. Lei-Ke Zhang; Wuhan Institute of Virology, CAS) were cultured in Dulbecco's modified Eagle medium (DMEM, Gibco, Grand Island, NY, USA) supplemented with 10% FBS. BSR-T7 cells were treated with 1 mg/mL G418 (Invitrogen) at every other passage [26]. All the cells above were cultured at 37°C with 5% $CO_2$. HEK293F cells (Thermo) were propagated in Freestyle 293 expression medium (SinoBiological, China) at 37°C with 8% $CO_2$, shaking at 180 rpm/min.

The CCHFV strain YL16070 (GenBank accession numbers KY354080, KY354081, KY354082) used in this study was stored in -80°C at the National Virus Resource Center (NVRC, IVCAS 6.6329). The viral titer (median tissue culture infective dose, $TCID_{50}$/mL) was determined using an indirect immunofluorescence assay (IFA) in Vero E6 cells [27]. All experiments on infectious CCHFV were conducted in a BSL-3 laboratory.

### Expression and purification of CCHFV Gc proteins

The Gc proteins (sGc-trimer) containing the ectodomain of the CCHFV IbAr10200 strain Gc (residues 1049–1569 of the full-length M segment, GenBank accession number AAM48106) were generated as previously described [15]. Summarily, the *Drosophila* S2 cells stably expressing sGc-trimer were induced using copper sulfate, and the supernatant was collected and clarified by centrifugation at 4°C, followed by purification with Ni-NTA agarose (Roche) and Superdex-200 column (GE Healthcare) equilibrated in phosphate buffered saline (PBS). The

purified sGc-trimer proteins were concentrated to 2 mg/mL and stored at -80˚C before the immunization of mice.

The Gc ectodomain (residues 1062–1582 of the full-length M segment; GenBank accession number AQX34599.1) of CCHFV-YL16070 was cloned into a pHCMV vector containing an S-tag at the N-terminus, and the recombinant protein, termed sGc-S-tag, was expressed in HEK 293F cells and purified as described in a previous study [14]. sGc-S-tag proteins were screened for mAbs using enzyme-linked immunosorbent assay (ELISA).

## Preparation of monoclonal antibodies

Six- to eight-week-old BALB/c mice were inoculated with the prepared sGc-trimer proteins emulsified with Freund's complete adjuvant. The immune response was boosted by three additional injections of sGc-trimer proteins emulsified with Freund's incomplete adjuvant at two-week intervals. Two weeks later, the mice were inoculated with sGc-trimer proteins without adjuvants for the final immunization. Three days later, splenocytes from the mice were fused with SP2/0 myeloma cells. After monoclonal screening, 37 hybridoma cells were selected and injected into the mice to prepare the ascites. The IgG antibodies in the supernatant of the ascites were purified using the caprylic acid-ammonium sulfate precipitation method and stored at -80˚C before use. The concentration of mAbs was determined using the Bradford Protein Assay Kit (Beyotime, China) by measuring the absorbance at 595 nm using a microplate reader (BioTek Instruments). The antibody purity was assessed via coommassie brilliant blue staining using SDS-PAGE.

## Generation of pseudotyped CCHFV

The pseudotyped CCHFV was generated as described previously [28]. Briefly, CCHFV-I-bAr10200 GP (GenBank accession number **AAM48106**) with C-terminal 53 amino acid deletion was cloned into the pCAGGS vector to generate pCAGGS-CCHFV-GPdel53aa, and the plasmid was verified through sequencing. HEK 293T cells in six-well plates were transfected with pCAGGS-CCHFV-GPdel53aa plasmids, followed by infection with *G-VSVΔG/GFP at a multiplicity of infection (MOI) of 1 for 3 h. Subsequently, the cell monolayer was washed with PBS 3 times, replaced with 10% FBS DMEM, and further incubated at 37˚C with 5% $CO_2$. After 24 h, the supernatant was collected, clarified by centrifugation at 500 $g$, and sterilized through a 0.22 μm filter. The pseudotyped virus was titrated in Vero E6 cells by an end-point dilution assay and stored at -80˚C.

## Neutralizing antibody screening by pseudotyped CCHFV

The pseudotyped CCHFV of 3,000 $TCID_{50}$ in a total volume of 200 μL EMEM medium containing 2% FBS was added to an equal volume of serially-diluted hybridoma supernatant, or EMEM medium as control. After 1 h incubation at 37˚C, a 100 μL virus-antibody mixture was added onto the Vero E6 cell monolayers in 48-well cell culture plates and incubated for an additional 1 h at 37˚C. Afterwards, the medium was discarded, washed 3 times by PBS, and replaced with 200 μL fresh EMEM. At 24 h p.i., cells expressing GFP were counted under a fluorescence microscope (EVOS FL Auto, Life Technologies) and the inhibition rates were calculated.

## Neutralization assay by live CCHFV

Live CCHFV-YL16070 was incubated with serially-diluted mAbs for 1 h at 27˚C, and then the virus-antibodies mixture was applied to Vero E6 cells pre-seeded in 48-well plates. After

adsorption for 1 h at 37˚C in a humidified incubator, the mixture was discarded, washed 3 times with PBS, and replaced with fresh EMEM. At 48 h p.i., supernatants were collected for titer determination and RNA extraction using a TaKaRa MiniBEST Viral RNA/DNA Kit (Cat. No. 9766) and viral genomic copies were determined by quantitative real-time PCR (qRT-PCR) with forward primer: 5′-TCAAGTGGAGGAAGGACATAGG-3' and reverse primer: 5′-TCCACATGTTCACGGCTCACTGGG-3'. The % inhibition = (1- sample copies (titer)/negative control copies (titer)) × 100%, and the median inhibitory concentration (IC$_{50}$) values were calculated using non-linear regression analysis, as described in a previous study [29]. The infected cells were fixed with 4%paraformaldehyde (PFA) overnight and used to detect viral nucleoproteins (NP) by immunofluorescence staining using an anti-NP mAb, as described in a previous study [30].

### Animal protection assay

All animal experiments were conducted with the approval of the Institutional Animal Care and Use Committee (IACUC) of the Wuhan Institute of Virology, Chinese Academy of Sciences, in Animal Biosafety Level 3 (ABSL-3) facilities (Ethics number: WIVA33202005). Eight to thirteen-week-old female type I interferon receptor knockout mice (C57BL/6J-IFNAR$^{-/-}$) were used. Mice were intraperitoneally infected with a 5,000 TCID$_{50}$ dose of CCHFV-YL16070. Mice were intraperitoneally injected with a 50 mg/kg dose of Gc8, Gc13, and Gc35 one day before viral infection. Mice injected with an equal volume of PBS were used as controls. The mice were monitored daily for physical condition and body weight, and were euthanized at the ethical endpoint (body weight loss reaching 20% of starting weight).

### Western blot for epitope mapping

To map the epitopes of the Gc antibodies, four truncated Gc fragments (aa 1–142, 132–262, 252–375, and 365–521) of CCHFV-IbAr10200 were PCR-amplified and cloned into the pET32a prokaryotic expression plasmid (Novagen). Protein samples were subjected to SDS-PAGE and transferred onto a polyvinylidene difluoride (PVDF) membrane. Subsequently, the membrane was blocked with 5% non-fat dried milk for 1 h at 25˚C, followed by incubation with Gc8, Gc13, or Gc35 for 2 h at 1:5,000 dilution. After washing with TBS-T (20 mM Tris-base, 150 mM NaCl, 0.05% Tween-20, pH 7.5), the blots were incubated with HRP-conjugated Affinipure Goat Anti-Mouse IgG (H+L) (Proteintech, 1:5,000 dilution in PBS-T) for 1 h. Final signals were detected using Pierce ECL Plus Western Blotting Substrate (Thermo Scientific) and photographed under MicroChemi (DNR Bio-Imaging Systems, DNR).

### Immunofluorescent staining

To evaluate whether these antibodies recognized native Gc proteins, immunofluorescence staining was performed. Vero E6 cells in 96-well plates were infected with live CCHFV-YL16070 at an MOI of 1. At 72 h p.i., the cells were fixed with 4%polyformaldehyde overnight, permeabilized with 0.2% Triton X-100, and stored at -80˚C before detection of Gc proteins. The plates were blocked with 5% bovine serum albumin (BSA) in PBS for 1 h and then incubated with Gc8, Gc13, or Gc35 at a dilution of 1:5,000 for 2 h, followed by rinsing with PBS and incubation with goat anti-mouse Alexa Fluor 488 (ab150113, 1:2,000 dilution) for 1 h. Nuclei were stained with Hoechst 33258 (Cat no. C1018, Beyotime, China). Images were obtained using a fluorescence microscope (EVOS FL Auto; Life Technologies).

## Competitive enzyme-linked immunosorbent assay (ELISA)

Competitive ELISA was performed to classify Gc8, Gc13 and Gc35, competitive ELISA was carried out. Firstly, an ELISA antigen plate was coated with 1 μg/mL purified sGc-S-tag protein and then incubated with 100 μL serially diluted biotin-labeled Gc8, Gc13, or Gc35 at 37˚C for 1 h, followed by addition of 100 μL avidin-HRP (Thermo Scientific) at 37˚C for 30 min. The mixture was discarded and the plate washed with PBS 5 times. Subsequently, tetramethylbenzidine (TMB) and hydrogen peroxide ($H_2O_2$) were added into the plate to react for 10 min at room temperature, avoiding exposure to light. Afterwards, $H_2SO_4$ was added to terminate the reaction. Optical density (OD) values between 1.5 and 2.0 were selected, and the corresponding dilution of the biotin-labeled antibody was used for further experiments. For formal competitive experiments, 50 μL biotin labeled antibodies with selected dilution and 50 μL serially-diluted, unlabeled antibodies were mixed and added to the Gc protein-coated ELISA plate instead of 100 μL biotin labeled antibodies, and the follow-up methods were the same as above. The OD values were determined and graphed using GraphPad Prism 9.0 (GraphPad, Inc., La Jolla, CA, USA).

## Biolayer interferometry

The binding characteristics of mAbs to sGc-S-tag were measured by Biolayer interferometry (BLI) using an Octet-Red 96 instrument (Pall ForteBio LLC, CA, USA). Summarily, sGc-trimers were labeled with biotin at a ratio of 1:30 at room temperature for 30 min, followed by dialysis with PBS to remove residual biotin. Then the labeled Gc proteins were diluted to a final concentration of 25 μg/mL in PBS containing 0.1% BSA and 0.02% Tween-20 to avoid nonspecific adsorption. Streptavidin biosensors (ForteBio) were soaked in the same buffer for 100 s to acquire the baseline before being moved to Gc-biotin for 300 s to complete the loading procedure. The sensors were then dipped into a buffer (180 s) to wash away the non-immobilized Gc-biotin. For the association step, the sensors were soaked in different concentrations of mAbs for 600 s, followed by dissociation in a buffer for 600 s. The binding curves and kinetics of association ($K_{on}$) and dissociation ($K_{off}$) were analyzed using the ForteBio Data Analysis Software.

## Fusion inhibition assay

Huh-7 cells were transfected with a pT7-eGFP reporter plasmid that could transcribe eGFP RNA using T7 RNA polymerase. BSR-T7/5 cells stably expressing T7 polymerase were transfected with the pCAGGS-CCHFV-GPdel53aa plasmid. The medium was replaced by pH 5.5 DMEM (acidified with HCl) at 24 h p.t. and incubated at 37˚C for 20 min in a humidified incubator. Huh-7 cells were detached using trypsin, mixed with BSR-T7/5 cells, and co-cultured in fresh DMEM (control group) or different concentrations of mAbs (experimental group). After 24 h, the co-cultured cells were fixed with 4% PFA overnight, and the nuclei were stained with Hoechst 33258 (Cat no. C1018, Beyotime, China). Images were obtained using a fluorescence microscope (EVOS FL Auto; Life Technologies). The total number of cells and syncytia were counted using ImageJ software, and the ratio of syncytia to total cells was calculated. The ratio in the experimental group was normalized to that in the control group. Statistical analyses were performed using an unpaired t-test (**$P < 0.01$, ***$P < 0.001$).

## Cryo-EM sample preparation

To obtain Fab fragment, Gc8 and Gc13 mAbs were digested with papain at 1:200 (w/w) ratio in 1×PBS buffer (supplemented with 10 mM L-cysteine and 2 mM EDTA) at 37˚C overnight.

Gc8 Fab and Gc13 Fab were purified by affinity chromatography using protein A resin, following the manufacturer's protocols (TransGen Biotech), and then by size-exclusion chromatography with a Superdex 200 (10/300) GL column (GE Healthcare) equilibrated with PBS.

The CCHFV sGc-trimer was purified as described in a previous study [15]. The purified sGc-trimer was incubated with Gc8 or Gc13 Fab in a 1:3.5 molar ratio on ice for 20 min. The complex samples were examined by negative-staining electron microscopy. The samples (0.02 mg/mL) were incubated for 4 min on glow-discharged grids with carbon support film and subsequently stained with 1% uranyl acetate solution. The grids were imaged using Talos L120C (Thermo Fisher Scientific) equipped with a CETA 16M detector. Further, 4 μL of sample solutions (0.3 mg/mL) were applied onto freshly glow-discharged holey carbon grids (Cu 200 mesh R1.2/1.3, Quantifoil), and the grids were vitrified using a Vitrobot Mark IV (Thermo Fisher Scientific) with a blot force of 5 for 3 to 4 s at 6°C and 100% humidity.

### Cryo-EM data collection and image processing

Cryo-EM data were acquired using a CRYO ARM 300 electron microscope (JEOL) operated at 300 kV with an in-column energy filter (slit width 20 eV). The movies were recorded on a K3 direct electron detector (Gatan) by the serial EM software [31] at a nominal magnification of 50,000× with a super-resolution pixel size of 0.475 Å and a defocus range of 0.5–2.5 μm. Each movie was dose-fractionated into 40 frames with an accumulated exposure of ~40 e$^-$/Å$^2$.

After manual selection, 2,367 and 3,344 movies of Gc8–Gc and Gc complexes, respectively, were selected. The recorded movies were binned and motion-corrected using MotionCor2 implemented in Relion [32]. Dose-weighted images were imported into cryoSPARC [33] to estimate contrast transfer function (CTF) parameters, particle picking, and extraction. Three rounds of reference-free 2D classification were performed using the Cryo-SPARC software. Particles with well-defined 2D class averages were selected for three *Ab initio* reconstructions, followed by heterogeneous and non-uniform (NU) refinements. To improve the density of the Fab–Gc complex interface, local refinement was performed using a soft mask encompassing the Gc8 or Gc13 variable domains and the fusion loops of Gc. Local refinement yielded reconstructions of Gc and Gc13–Gc complex structures at 3.1 Å and 3.0 Å resolution, respectively, based on the gold-standard Fourier shell correlation (FSC) of 0.143 criterion **S1 Table**).

### Model building and refinement

Our previously reported sGc-trimer structure (PDB: 7FGF) was docked into the cryo-EM maps using UCSF Chimera [34] and the variable domains of the light and heavy chain (VL and VH) were manually built. Iterative rounds of manual adjustment and automated rebuilding were carried out using Coot [35] and Phenix [36], respectively. The final models were validated using Molprobity [37] and the statistics are summarized in **S1 Table**. Figures were generated using the UCSF ChimeraX [38] and PyMOL (PyMOL Molecular Graphics System, version 1.8; Schrödinger, LLC). The cryo-EM maps and atomic models were deposited into the Electron Microscopy Data Bank and PDB with accession codes EMD-36368 and 8JKD (sGc-trimer), EMD-36406 and 8JLW (local reconstruction of Gc8 in complex with the sGc-trimer), and EMD-36407 and 8JLX (local reconstruction of Gc13 in complex with the sGc-trimer).

### Supporting information

**S1 Fig. Body weight curve of C57BL/6J-IFNAR$^{-/-}$ mice administered with Gc8, Gc13 and Gc35.**
(PDF)

**S2 Fig.** Representative negative staining micrographs of CCHFV Gc (A), the Gc-Gc8 complex (B), and the Gc-Gc13 complex (C). Scale bar: 50 nm.
(PDF)

**S3 Fig. Cryo-EM data processing of Gc-Gc8 complex:** representative micrograph (A), 2D class averages (B), 3D classes (C), and local refinement of Gc-Gc8 interface(D).
(PDF)

**S4 Fig. Cryo-EM data processing of Gc-Gc13 complex:** representative micrograph (A), 2D class averages (B), 3D classes (C), and local refinement of Gc-Gc13 interface(D).
(PDF)

**S5 Fig. CCHFV M segment based phylogenetic analysis.** Phylogenetic tree was analyzed using MEGA version 6.0, distances were calculated by Kimura's 2-parameter and a phylogenetic tree was plotted by the neighbor-joining method based on full segment of M. Thirty-three isolates were classified into eight clades.
(PDF)

**S1 Table. Cryo-EM data collection and refinement statistics.**
(PDF)

**S2 Table. Interactions between Gc8 light chain variable (VL) region and Gc.**
(PDF)

**S3 Table. Interactions between Gc8 heavy chain variable (VH) region and Gc.**
(PDF)

**S4 Table. Interactions between Gc13 light chain variable (VL) region and Gc.**
(PDF)

**S5 Table. Interactions between Gc13 heavy chain variable (VH) region and Gc.**
(PDF)

**S6 Table. Residues/atoms involved in hydrogen bonds between Fab and the fusion loops.**
(PDF)

**S7 Table. Fusion loop sequences of different CCHFV isolates.**
(PDF)

## Acknowledgments

We thank Jia Wu, Hao Tang, and Jun Liu from the BSL-3 Laboratory; Xuefang An, Fan Zhang, Yuzhou Xiao, He Zhao, and Li Li from the animal center of Wuhan Institute of Virology, and Dr. Ding Gao from the Centre for Instrumental Analysis and Metrology of the Wuhan Institute of Virology.

## Author Contributions

**Conceptualization:** Sheng Cao, Manli Wang.

**Data curation:** Liushuai Li, Tingting Chong, Lu Peng.

**Formal analysis:** Liushuai Li, Tingting Chong.

**Funding acquisition:** Jia Liu, Zhihong Hu, Manli Wang.

**Investigation:** Liushuai Li, Tingting Chong, Lu Peng, Yajie Liu, Guibo Rao, Yan Fu, Yanni Shu, Jiamei Shen, Qinghong Xiao, Jia Liu.

**Methodology:** Liushuai Li, Tingting Chong, Lu Peng, Bing Yan, Sheng Cao, Manli Wang.

**Resources:** Jiang Li, Fei Deng.

**Supervision:** Zhihong Hu, Sheng Cao, Manli Wang.

**Writing – original draft:** Liushuai Li, Tingting Chong.

**Writing – review & editing:** Zhihong Hu, Sheng Cao, Manli Wang.

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
