## [Decision Letter · Decision Letter 0]

14 Oct 2023

Dear Dr. Wang,

Thank you very much for submitting your manuscript "Neutralizing monoclonal antibodies against the Gc fusion loop region of Crimean–Congo hemorrhagic fever virus" for consideration at PLOS Pathogens. As with all papers reviewed by the journal, your manuscript was reviewed by members of the editorial board and by several independent reviewers. In light of the reviews (below this email), we would like to invite the resubmission of a significantly-revised version that takes into account the reviewers' comments.

We cannot make any decision about publication until we have seen the revised manuscript and your response to the reviewers' comments. Your revised manuscript is also likely to be sent to reviewers for further evaluation.

Sincerely,

Amy L. Hartman, PhD

Academic Editor

PLOS Pathogens

Matthias Schnell

Section Editor

PLOS Pathogens

Kasturi Haldar

Editor-in-Chief

PLOS Pathogens

orcid.org/0000-0001-5065-158X

Michael Malim

Editor-in-Chief

PLOS Pathogens

orcid.org/0000-0002-7699-2064

Reviewer's Responses to Questions

**Part I - Summary**

Reviewer #1: The manuscript by Li et al. identify several nAbs from mice, evaluate their neutralization capacity, therapeutic efficacy in mice and structurally investigate how these nAbs bind to CCHFV. Overall, the therapeutic efficacy of the identified mAbs is modest, however, authors acknowledge this. Authors mechanistic investigations of the nAb binding further highlight the epitopes on CCHFV glycoproteins that are targeted by nAbs. I would encourage authors to expand on their discussion to speculate on how they think their data predicts the therapeutic value of nAbs for CCHFV.

Reviewer #2: In this study, the authors identified two antibodies neutralizing infection by the Crimean Congo hemorrhagic fever virus, an emerging virus with a high pandemic risk. To this extent, the results shown here are of high biological significance. Having written this, the study remains highly descriptive. The results support the authors' conclusion. The experiments are, on the whole, well-controlled. The manuscript would benefit from proofreading in English style. I have a few more specific major and minor points (see below).

**Part II – Major Issues: Key Experiments Required for Acceptance**

Reviewer #1: Major comments

Line 90-98: This section is confusing and although the methods clarify it somewhat, this results section needs to be clarified. How the mice were immunized and with what should be mentioned in the results to indicate how the hybridomas were generated. Line 96 – 99, what does this mean? Were hybridomas injected into mice or were antibody supernatants injected into mice? Why were mAbs generated from ascites instead of traditional in vitro production systems?

Line 115: How were antibodies quantified? How was purity assessed?

Line 147-151: Authors show improved survival but no difference in viral loads in the liver or spleen. Although authors state that the mean was reduced by one log, 5 of 7 control treated animals had lower viral loads in the liver than the mean of Gc13 treated animals, yet 100% of control treated animals died versus only 3 of the 8 Gc13 treated animals. Is protection from death due to viral control in other tissues? Blood? Brain? Authors may also want to highlight that they used a high dose challenge (5000 TCID50) in IFNAR-/- mice when the LD50 for CCHFV in IFNAR mice has been reported to be as low as 0.06 TCID50 (Hawman et al. 2018 Antiviral Research or Zivcec et al. 2013 JID). Although those LD50s are with other strains of CCHFV, authors may have challenged mice with ~100,000 LD50s. This may also contribute to only partial protection and also suggests that even the level of protection authors observed is notable. Authors should compare this to the challenge model used in the human nAb manuscript.

Line 333-337: Could it also be due to the immunogen used in author’s study? How sure are the author’s that their Gc antigen was properly folded? It is possible that since the abs from convalescent survivors arose in response to authentic antigen whereas the authors abs arose from recombinant protein that the recombinant protein was improperly folded.

The discussion should wrap up with a conclusion paragraph. In the discussion I would encourage author’s to speculate on how they think their data predicts the therapeutic potential of nAbs for CCHFV. Author’s data adds to an overall modest efficacy of nAbs for CCHFV. Author’s saw partial protection even with pre-treatment and the referenced paper using nAbs from human survivors also saw incomplete protection as soon as mAbs were evaluated therapeutically. How does this picture compare to nAbs for other hemorrhagic fevers such as Ebola? Have authors considered screening non-neutralizing nAbs for protective efficacy?

Reviewer #2: 1. In Fig. 2A, the authors assessed the viral RNA but no information is provided about the release of infectious progeny.

2. Fig. 2B without any quantification is not informative. It's hard to see any difference by eye in these series of fluorescence microscopy images.

3. Fig. 3 missed non-infected mice, which would make it possible to assess the effect of the antibodies on the survival of animals without the virus.

4. Similar to Point 2, Fig. 4B is not very informative as it is.

5. Fig. 6 is not convincing, essentially because it is not possible to see syncytia on IF pictures shown in 6C.

6. The TEM work shown in Figures 7 and 8 is too succinctly described. In general, the Results section lacks information and description in many places.

7. The discussion is overall a retelling of the results.

**Part III – Minor Issues: Editorial and Data Presentation Modifications**

Reviewer #1: Abstract is too detailed, I would consider shortening it.

Line 56: clarify that it is the Nairoviridae family in the Bunyavirales order.

In figure 7 would recommend highlighting the fusion loop to make it clearer to the reader when author’s state the abs target the fusion loop. Particularly panel C?

Line 306: sulking should be suckling

Reviewer #2: 1. Line 34 (abstract), “confirmed” is inappropriate. “Showed”, “indicated” etc. would be better.

2. The Author Summary is too specialized and not accessible enough to a wide audience.

3. The Results section should begin with a brief explanation of the origin of hybridomas. This is done, but far too late in the discussion.

4. In the context of work on Gc-neutralizing antibodies, the first published structure for CCHFV Gc should be cited (PMID: 34793197).

5. The term “strains” to describe antibodies seems inappropriate to me.

6. In Fig. 5A, the term “naked” to describe the antibodies that are not biotinylated also seems inappropriate to me.

PLOS authors have the option to publish the peer review history of their article (what does this mean?). If published, this will include your full peer review and any attached files.

Reviewer #1: No

Reviewer #2: No
---

## [Editor Report · Decision Letter 1]

4 Jan 2024

Dear Dr. Wang,

We are pleased to inform you that your manuscript 'Neutralizing monoclonal antibodies against the Gc fusion loop region of Crimean–Congo hemorrhagic fever virus' has been provisionally accepted for publication in PLOS Pathogens.

Best regards,

Amy L. Hartman, PhD

Academic Editor

PLOS Pathogens

Matthias Schnell

Section Editor

PLOS Pathogens

Kasturi Haldar

Editor-in-Chief

PLOS Pathogens

orcid.org/0000-0001-5065-158X

Michael Malim

Editor-in-Chief

PLOS Pathogens

orcid.org/0000-0002-7699-2064
---

## [Editor Report · Acceptance letter]

23 Jan 2024

Dear Dr. Wang,

We are delighted to inform you that your manuscript, "Neutralizing monoclonal antibodies against the Gc fusion loop region of Crimean–Congo hemorrhagic fever virus," has been formally accepted for publication in PLOS Pathogens.

Best regards,

Michael Malim

Editor-in-Chief

PLOS Pathogens

orcid.org/0000-0002-7699-2064